# Enhanced Photoluminescence and Photocatalytic Efficiency of La-Doped Bismuth Molybdate: Its Preparation and Characterization

**DOI:** 10.3390/ma13010035

**Published:** 2019-12-20

**Authors:** Muhammad Waqar, Muhammad Imran, Syed Farooq Adil, Sadia Noreen, Shoomaila Latif, Mujeeb Khan, Mohammed Rafiq H. Siddiqui

**Affiliations:** 1Institute of Chemistry, Punjab University Lahore-Pakistan, Lahore 54590, Pakistan; waqar0100@gmail.com (M.W.); sadian543@gmail.com (S.N.); 2Department of Chemistry, King Saud University, Riyadh 11451, Saudi Arabia; kmujeeb@ksu.edu.sa (M.K.); rafiqs@ksu.edu.sa (M.R.H.S.); 3Department of Chemistry, University of Lahore-Pakistan, Lahore 54590, Pakistan; shoomaila_latif@yahoo.com

**Keywords:** lanthanides doping, bismuth molybdate, nanoparticles, luminescence, photo degradation, methylene blue

## Abstract

Herein, a systematic study of the enhanced physicochemical properties of lanthanide doped (La-doped) bismuth molybdate (Bi_2_MoO_6_) is performed. For this purpose, Bi_2_MoO_6_ and La-doped Bi_2_MoO_6_ were prepared by the sol-gel method. BiCl_3_, Na_2_MoO_4_^·^2H_2_O, and LaCl_3_·7H_2_O were taken as the main precursors while sodium dodecyl sulfate was used as a surfactant. Both Bi_2_MoO_6_ and La-doped Bi_2_MoO_6_ were calcined at 650 °C for 2 h. These prepared materials were characterized by spectroscopic techniques such as UV–VIS, FT-IR, XRD, photoluminescence, XPS, along with other techniques such as SEM, TEM, TGA, etc. The investigation of luminescence behavior revealed that the La-doped Bi_2_MoO_6_ nanocomposite exhibited much greater luminescence compared to the undoped Bi_2_MoO_6_. The photocatalytic behavior of the prepared materials was explored by studying the degradation of methylene blue (MB) at room temperature. The degradation of MB with Bi_2_MoO_6_ and La-doped bismuth molybdate were observed to be 68% and 75% @ 45 s, respectively, indicating an enhancement of catalytic performance due to the La doping.

## 1. Introduction

Light emitting diodes with higher efficiency (>100,000 h) are a potential solution for the growing needs of urban civilization. Among various materials applied for the preparation of LEDs, indium gallium nitride (InGaN), aluminum gallium nitride (AlGaN) have been found to be more efficient as LEDs. Additionally, various other materials have also been applied for this purpose, such as, borates [1], phosphors [2], MOFs [3], different types of mixed metal oxides including CeO_2_, ZnO, ZnAl_2_O_4_ [4], Cr-doped ZnGa_2_O_4_ [5]. In recent times, scientists have paid more attention towards the preparation and characterization of trivalent lanthanide doped materials due to their applicability as solid-state laser media, fiber amplifiers, infrared to-visible up-converters, field emission displays [6,7] and, therefore, numerous luminescent materials were synthesized with suitable rare earth activators [Ln^3+^].

Nanomaterials have gained great importance due to remarkable physicochemical properties such as, catalytic, dielectric and luminescence. These properties can be further enhanced by altering the morphology of the nanomaterials. Among the plethora of nanomaterials, a series of Bi-ion based materials such as Bi_2_O_3,_ BiOBr, Bi_2_S_3_, BiVO_4_, Bi_2_WO_6_, Bi_2_Ru_2_O_7−__d_, Bi_2_Ti_2_O_7_, Bi_12_TiO_20_, and Bi_2_InNbO_7_ have attracted lot of attention as photocatalysts that operate under visible range [8,9,10,11,12] and have been reportedly employed for a variety of applications such as photocatalytic decomposition of water, and degradation of harmful dyes from water bodies [13,14,15], as almost 15% of the dyes used are released in water, which have adverse effects on the environment [16,17].

Apart from bismuth, molybdenum oxide (Mo)- and tungsten (W)-based compounds were also reported as efficient materials in the field of photo-catalysis and other important applications, such as photoluminescence, optical fibers, and microwave applications, etc. [18,19]. In this regard, the combination of Bi with Mo and W, such as, Bi_2_MO_6_ (M = Mo, W) types of materials have great potential as efficient photocatalysts in the visible solar spectrum for the degradation of organic compounds and water spitting [20,21]. Particularly, bismuth molybdate (Bi_2_MoO_6_)-based photocatalysts exhibit several special properties among other types of metal molybdates for the degradation of organic compounds, especially dyes. So far, the Bi_2_MoO_6_ nano-structure has been prepared through various conventional procedures, such as co-precipitation, hydrothermal, solid state reaction and sol-gel methods [20].

As it is well documented in literature that the incorporation of metals and non-metals with metal oxides enhances the catalytic efficiency [22,23]. In this research work, we employ a simple and facile sol-gel method to synthesize Bi_2_MoO_6_ and La-doped Bi_2_MoO_6_ nanocomposites using sodium dodecyl sulfate as a surfactant. The sol-gel method used in the present research work has a number of advantages, such as the control of the morphology and being a low-temperature process. The prepared materials were subjected to thorough characterization, such as SEM, TEM, BET, FT-IR, TGA, XPS, XRD, and photoluminescence spectra. The prepared materials were employed as photo-degradation catalysts for the degradation of methylene blue, an environmentally hazardous dye.

## 2. Materials and Methods 

### 2.1. Materials

All chemicals used for the synthesis such as, sodium molybdate (Na_2_Mo_2_O_4_·2H_2_O), bismuth chloride (BiCl_3_), and lanthanum chloride (LaCl_3_·7H_2_O), were of analytical grade purchased from Sigma Aldrich and PubChem (Saint Louis, USA).

The Bi_2_MoO_6_ was synthesized by the following method. Sodium dodecyl sulfate (0.576 g, 2 mmol) was dispersed into deionized water with constant stirring and temperature was elevated up to 50 °C. At this temperature sodium molybdate (0.483 g, 2 mmol) was added drop wise followed by drop wise addition of bismuth chloride (0.63 g, 2 mmol) to from sol with continuous stirring. The resulting mixture was stirred for further 2 h, while the temperature was raised to 90 °C. After that the solution was filtered and washed with deionized water and ethanol. The obtained precipitate was dried in oven at 60 °C for 3 h. The dried precipitate was ground into powder by mortar and pestle followed by calcination at 650 °C in furnace for 2 h.

La-doped Bi_2_MoO_6_ catalyst was synthesized in a similar manner as described above with a slight modification. The sodium dodecyl sulfate (0.576 g, 2 mmol) was dispersed into deionized water at 50 °C, to this solution sodium molybdate (0.483 g, 2 mmol) was added drop-wise followed by the drop-wise addition of lanthanum chloride (0.742 g, 2 mmol) which yielded a white suspension. To this, bismuth chloride (0.63 g, 2 mmol) was added drop-wise and the resulting mixture was heated at 90 °C under continuous stirring for 2 h which resulted in the formation of yellow precipitate. The obtained precipitate was dried in oven at 60 °C for 3 h. The dried precipitate was ground into powder with a mortar and pestle followed by calcination at 650 °C in furnace for 2 h.

### 2.2. Characterization Techniques 

The characterizations of synthesized Bi_2_MoO_6_ and La-doped Bi_2_MoO_6_ were carried out by using different analytical techniques to understand their morphology and other catalytic activities. The functional groups in Bi_2_MoO_6_ and La-doped Bi_2_MoO_6_ were determined by Agilent Technologies Cary 630 Fourier transform infrared (FT-IR) spectrometer (Agilent, Santa Clara, USA). Degradation of dye was recorded by T90+ UV–VIS spectrometer (Perkin Elmer lambda 35, Waltham, MA, USA). The photoluminescence (PL) properties of synthetic materials were analyzed by FLS 1000 photoluminescence spectrometer. BET surface area (NOVA 4200e surface area and pore size analyzer (Quantachrome Instruments, FL, USA)). The XRD analysis of the as-prepared nanocatalyst was carried out using a D2 Phaser X-ray diffractometer (Bruker, Bremen, Germany) with Cu Kα radiation (k = 1.5418 A°). XPS spectra were measured on a PHI 5600 Multi-Technique XPS (Physical Electronics, Lake Drive East, Chanhassen, MN, USA) using monochromatized Al Kα at 1486.6 eV. Peak fitting was performed using Origin software (2019a), USA. TGA was carried out using a TGA/DSC 1 (Mettler Toledo AG, Analytical, Schwerzenbach, Switzerland).

### 2.3. Photocatalytic Activity Measurements 

The catalytic activities of prepared Bi_2_MoO_6_ and La-doped Bi_2_MoO_6_ catalysts were determined by the degradation of methylene blue (MB) as model reaction in the presence of reducing agent NaBH_4_. We designed three reactions for degradation of MB. The time dependent degradation of methylene blue was examined from decrease in the absorbance observed in the UV–VIS spectra, scanned in the range of 400–800 at room temperature. To evaluate the catalytic activity of the prepared material (a) 5 mL solution of methylene blue was mixed with NaBH_4_ (0.2 mL, 0.16 M); (b) same solution in the presence of 2 mg of Bi_2_MoO_6_ catalyst at room temperature; and (c) the same solution in the presence of 2 mg of La-doped Bi_2_MoO_6_ catalyst at room temperature. The blue color of methylene blue disappeared with time which shows the degradation of methylene blue. The reduction efficiency and degradation were determined by the following equation:$$\mathrm{Reduction}\;\mathrm{efficiency}\left(\%\right)=\frac{\left(C_{0}-C_{t}\right)}{C_{0}}\times 100$$
where *C*_0_ and *C*_t_ represent the initial and residual concentration of methylene blue in solution with time. The reduction of methylene blue followed the reaction mechanism in which electrons passes from the reductant BH_4_^−^ to methylene blue which contains double bond. The Bi_2_MoO_6_ and La-doped Bi_2_MoO_6_ help to decrease the potential difference between donor and acceptor and facilitate the transfer of electron from BH_4_^–^ to MB for the hydrogenation of double bond present in methylene blue.

## 3. Results and Discussion

### 3.1. Microscopic Analysis

Figure 1 shows the SEM micrograms of the prepared Bi_2_MoO_6_ and La-doped Bi_2_MoO_6_ nanocomposite. The Bi_2_MoO_6_ surface morphology appears to be dense rod-like structures, however, the surface morphology of La-doped Bi_2_MoO_6_ is found to be globular and hemispherical particles are engaged coarse granular clustering with high porosity. The TEM images of the prepared Bi_2_MoO_6_ and La-doped Bi_2_MoO_6_ nanocomposite and their particle size distribution graph is given Figure 2. From the TEM images and the particle size distribution graph (Figure 2a,c) of Bi_2_MoO_6,_ it is evident that the nanoparticles range from 1–8 nm with a mean size of 3.5 nm, while that of La-doped Bi_2_MoO_6_ nanocomposites have a size range of 1–7 nm with a mean size of 3 nm (Figure 2b,d), indicating the effect of doping on the particle size of the composite.

### 3.2. BET Analysis

The synthesized Bi_2_MoO_6_ and La-doped Bi_2_MoO_6_ nanocomposites were subjected to BET analysis to understand the porosity and it was found to be mesoporous in nature. The surface area of the Bi_2_MoO_6_ and La-doped Bi_2_MoO_6_ nanocomposite was found to be 1.0 m²/g and 3.4 m²/g, respectively. The pore distribution plot revealed that the surface porosity for both the prepared material is different, the Bi_2_MoO_6_ possessed a pore volume in the range of 0.3–2.4 × 10^−3^ cm^3^/g, while the La-doped Bi_2_MoO_6_ nanocomposite was found to possess a pore volume in the range of 0.6–3.8 × 10^−3^ cm^3^/g, which is mostly represents granular materials [22]. The pore distribution plot is given in Figure 3.

### 3.3. FT-IR Spectral Analysis

FT-IR spectral data provided valuable information about functional groups present in synthesized Bi_2_MoO_6_ and La-doped Bi_2_MoO_6_ nanocomposites. The FT-IR spectra of Bi_2_MoO_6_ and La-doped Bi_2_MoO_6_ were recorded in the range 4000–400 cm^−1^. The spectrum of bismuth nanoparticle showed the peaks at 797 cm^−1^ is due to (Mo = O). The peak observed at 499 cm^−1^ was assigned to Bi-O bond [23]. The infrared spectrum of La-doped Bi_2_MoO_6_ was compared to that of un-doped bismuth molybdate and a shift was observed in the peaks of Mo-O and Bi-O from 797 cm^−1^ to 794 cm^−1^ and 499 cm^−1^ to 493 cm^−1^, respectively. This shift can be attributed to the influence of the dual lanthanum in the host matrix of Bi_2_MoO_6_ [24]. The IR spectra of Bi_2_MoO_6_ and La-doped Bi_2_MoO_6_ nanocomposite is given in Appendix A.

### 3.4. X-Ray Diffraction Spectroscopy 

The XRD pattern of the synthesized sample of bismuth molybdate (BW) is given in Figure 4a. It shows pronounced diffraction peaks of (131), (212), (260), (191) and (280) planes at 2-Theta = 28.271, 47.094, 47.214, 56.283, and 56.452, which can be ascribed to the characteristic orthorhombic phase of Bi_2_MoO_6_ (JCPDS card no. 77-1246). However, the lanthanum doped bismuth molybdate (LW) exhibits more than one crystal phases as shown in Figure 4b. It shows the main diffraction peaks of α-Bi_2_Mo_3_O_12_ at 2θ = 27.947°, 29.170°, 30.995°, and 48.348° corresponding to diffraction planes of (221), (023), (040) and (242), respectively, and match with JCPDS card no. 21-0103. It was found that the addition of La to the Bi_2_MoO_6_ induced a phase transition of synthesized bismuth molybdenum from γ-Bi_2_MoO_6_ to α-Bi_2_Mo_3_O_12_. The diffraction peaks of planes (110), (130), and (213) at diffraction angles 13.55, 30.98 and 37.57, respectively shows the presence of monoclinic crystal phase of lanthanum molybdenum oxide (LaMo_5_O_8_). Along with these two phases, the XRD pattern also shows the presence of bismuth oxide (Bi_2_O_3_) as presented by the diffraction peaks at 27.94, 46.20, and 54.24 correspond to (201), (222), and (203) planes. Moreover, the crystallite size of the Bi_2_MoO_6_ and La-doped Bi_2_MoO_6_ was calculated using Debye–Scherrer equation and the crystallite size obtained was found to be 18 and 14 nm. Normally the size of crystallites as obtained by XRD are smaller than the particle size obtained by TEM, as the particles are made up of crystallites. However, in our case the observation is opposite, this may be because XRD measurements are usually on bulk of the sample, whereas TEM is usually on a small selected area, which in our case showed smaller particle size.

### 3.5. Luminescence

The optical property of Bi^3+^ ions have been explained in terms of Russell–Saunders type electronic energy terms with s^2^ configuration in the ground and sp configuration (first excited state). Among these, the energetically lowest possible ^3^P_0_–^1^S_0_ excitation is strongly forbidden but the ^3^P_1_–^1^S_0_ transition is possible because of spin-orbit coupling of the ^3^P_1_ and ^1^P_1_ states [25]. Figure 5 represented the PL spectra of un-doped Bi2MoO6 and La-Bi_2_MoO_6_ in the range 350 to 750 nm excited by 350 nm [26]. The relatively broad PL peaks appeared around 700 nm with enhanced emission intensity in La-doped Bi_2_MoO_6_ compared to un-doped Bi_2_MoO_6_. PL intensity normally depends on crystallinity, materials having more crystallinity exhibits enhanced PL intensity. [27]. The same possible reason might be attributed to the enhanced emission intensity of La-Bi_2_MoO_6_. A similar observation has been reported for CaMoO_4_:Eu^3+^ in the literature [27].

### 3.6. X-Ray Photoelectron Spectroscopy

Surface composition analyses of the prepared nanocomposites were analyzed using X-Ray photoelectron spectroscopy (XPS). The survey scan revealed the presence of Bi, Mo, O elements in the composites prepared, and an additional signal corresponding to the element La revealed the successful doping of La in the nanocomposite given in Figure 6.

The deconvolution of the Bi 4f spectrum of Bi_2_MoO_6_ yielded peaks at 161.8 eV and 163.8 eV corresponding to Bi 4f_5/2_ and the difference in BE between the two Bi 4f_5/2_ peaks was 2.0 eV, which coincided with that between Bi^0^ and Bi^3^. While in the case of La-doped Bi_2_MoO_6_ nanocomposite the Bi 4f spectrum yielded peaks at 159.3 eV and 164.6 eV corresponding to Bi 4f_7/2_ and Bi 4f_5/2,_ respectively. However, the Mo 3d deconvoluted spectrum for the Bi_2_MoO_6_ yielded peaks at 234.6 eV and 236.7 eV attributed to Mo 3d_5/2_ which correspond to the Mo^6+^, while that of La-doped Bi_2_MoO_6_ nanocomposite yielded peaks at 233.2 eV, 236.7 eV, and 238.3 eV [28]. The deconvolution of the Bi 4f and Mo 3d spectrum of Bi_2_MoO_6_ and La-doped Bi_2_MoO_6_ nanocomposites are given in Figure 7.

### 3.7. TGA Analysis 

Thermogravimetric analysis (TGA) was conducted to identify the thermal behavior of the Bi_2_MoO_6_ and La-doped Bi_2_MoO_6_ nanocomposites and the results were compared as displayed in Figure 8. TGA of Bi_2_MoO_6_ reveals that there is an 8% weight loss when the sample is heated to 800 °C starting from 25 °C at a heating rate of 10 °C/minute under an inert atmosphere. However, when the La-doped Bi_2_MoO_6_ was subjected to similar study it is found that the weight loss is up to 16% indicating that the later nanocomposite is thermally less stable than un-doped Bi_2_MoO_6_.

### 3.8. Photocatalytic Activity

The catalytic activities of undoped Bi_2_MoO_6_ and La-doped Bi_2_MoO_6_ were evaluated by the degradation of methylene blue (MB) using UV-visible spectroscopy. λ-max for pure methylene blue was noted at 665 nm. The gradual change in color was observed at different time intervals which ensured the enhanced reduction. Maximum degradation was observed at a time interval of 45 s with complete decolorization. The presence of the reaction intermediates was eradicated as no new band appeared in the spectrum. The degradation of methylene blue using NaBH_4_ was recorded minimal 18% at 45 s, while in the presence of undoped BiMoO_6_ catalyst, the degradation of MB was enhanced up to 48% within first 15 s, as the reaction proceeds, the % age degradation was recorded as 55% and 68% at 30 and 45 s, respectively (Figure 9). However, when a similar reaction was carried out using NaBH_4_ in the presence of La-doped Bi_2_MoO_6_, a drastic increase in the degradation of methylene blue was observed, with a percentage of degradation of 53% in the first 15 s, which further proceeds to 69% and 75% at 30 and 45 s, respectively. The degradation kinetics of the reaction are illustrated graphically in Figure 10.

Relevant literature reveals that similar pattern of degradation has been observed with Pd-Nano-diamond-GO composite for the degradation of MB from 15 s to 60 s [23]. This enhancement of degradation efficiency can be attributed to the inclusion of La in the system. By correlating the characterization data obtained and the results of photocatalytic activity it can be concluded that the inclusion of La leads to the formation of nanocomposites with smaller dimension and larger surface area. These changes along with the phase transformation of γ-Bi_2_MoO_6_ to α-Bi_2_Mo_3_O_12,_ and the presence of monoclinic crystal phase of lanthanum molybdenum oxide (LaMo_5_O_8_) are assumed to be responsible in the enhancement of photocatalytic activity towards the degradation of MB.

## 4. Conclusion

In conclusion the Bi_2_MoO_6_ and La-doped Bi_2_MoO_6_ materials were successfully synthesized through the environmental friendly sol-gel method using sodium dodecyl sulfate as a surfactant. The morphological and structural properties of the synthesized material were determined by various analytical techniques. The morphology of Bi_2_MoO_6_ was changed with lanthanum doping, i.e., from dense rod-like structures to globular and hemispherical particles engaged into clusters with high porosity. The photo-degradation efficiency of La-Bi_2_MoO_6_ for MB was enhanced due to lanthanum doping and can be attributed to the efficient separation and preventing of the recombination of electron-hole pairs. Hence, the environmentally friendly sol-gel method for synthesis of La-Bi_2_MoO_6_ is an efficient tool for possible applications in the degradation of dyes and sequestering of MB pollutants from environment, and this study can be further extended to other pollutants, which is in progress and shall be published later.

## Figures and Tables

**Figure 1 materials-13-00035-f001:**
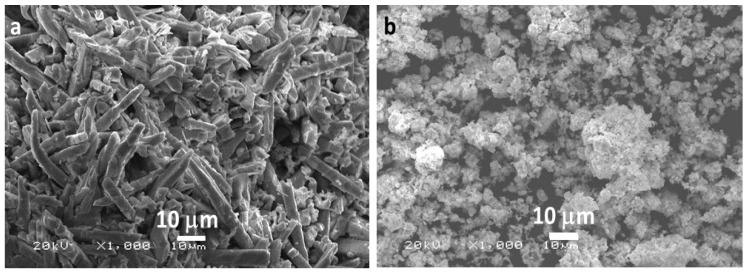
Scanning electron micrograms of (**a**) Bi_2_MoO_6_ and (**b**) La- Bi_2_MoO_6_ nanocomposite.

**Figure 2 materials-13-00035-f002:**
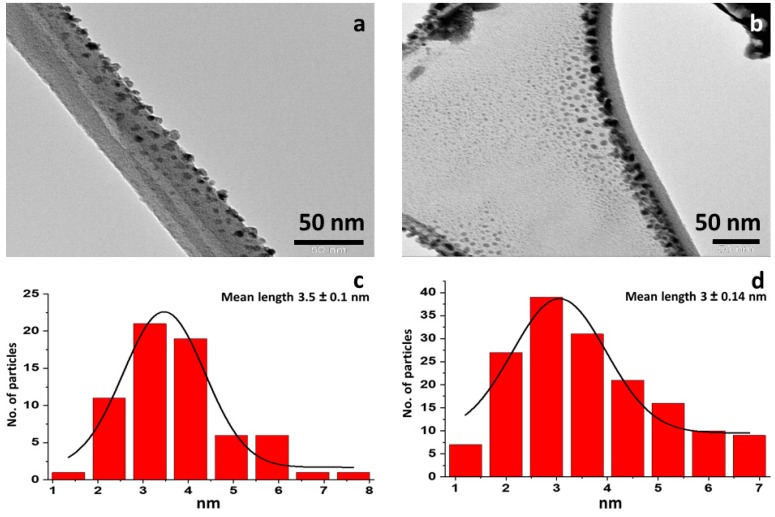
Transmission electron micrograms of (**a**) Bi_2_MoO_6_ and (**b**) La-doped Bi_2_MoO_6_ nanocomposite: Particle distribution graph of (**c**) Bi_2_MoO_6_, and (**d**) La-doped Bi_2_MoO_6_ nanocomposite.

**Figure 3 materials-13-00035-f003:**
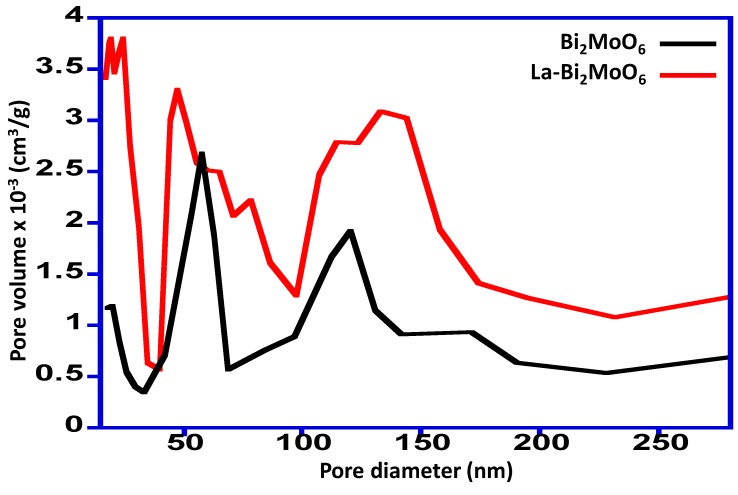
Pore size distribution curve of Bi_2_MoO_6_ and La-doped Bi_2_MoO_6_ nanocomposite.

**Figure 4 materials-13-00035-f004:**
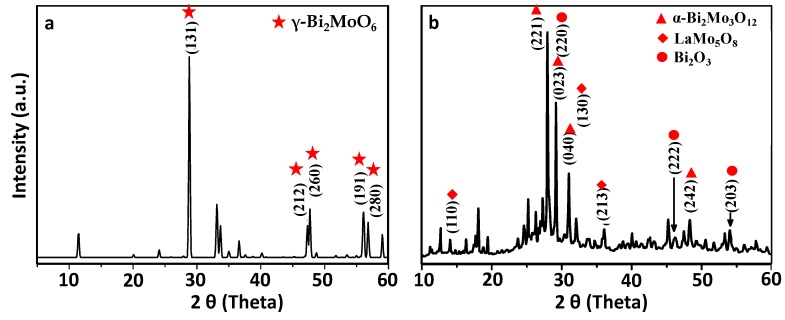
XRD pattern of (**a**) Bi_2_MoO_6_ and (**b**) La-doped Bi_2_MoO_6_ nanocomposite.

**Figure 5 materials-13-00035-f005:**
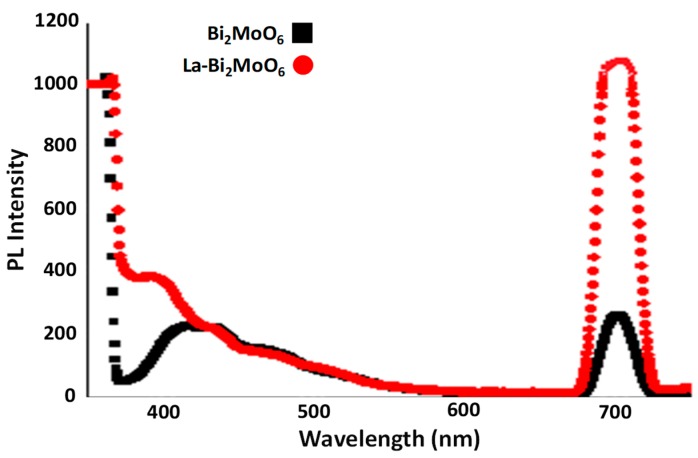
Photoluminescence (PL) spectra of Bi_2_MoO_6_ and La-doped Bi_2_MoO_6_.

**Figure 6 materials-13-00035-f006:**
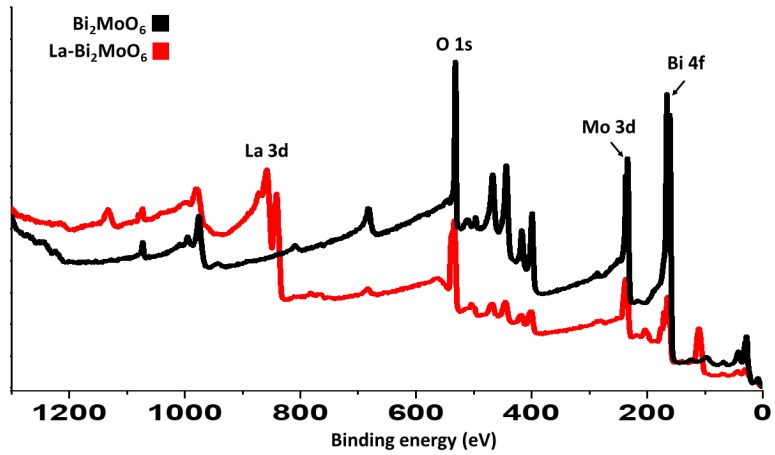
XPS survey scan of Bi_2_MoO_6_ and La-doped Bi_2_MoO_6_ nanocomposites.

**Figure 7 materials-13-00035-f007:**
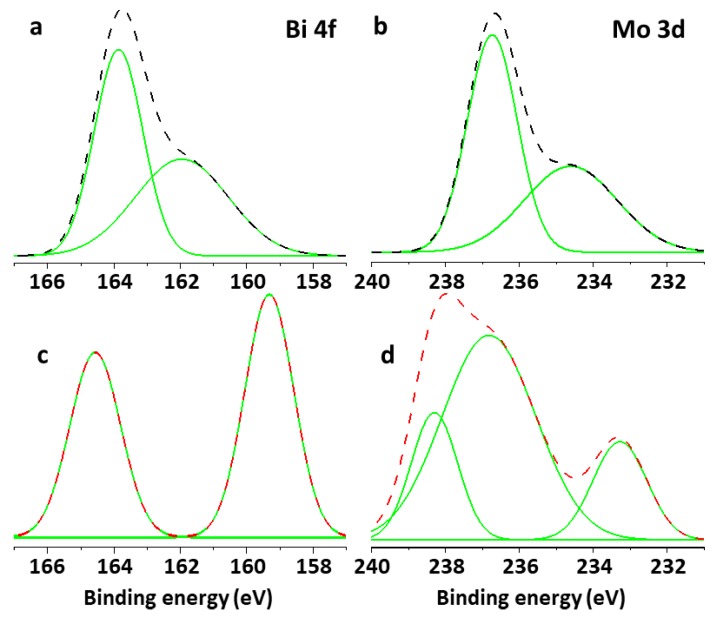
XPS results of deconvoluted Bi 4f region of (**a**) Bi_2_MoO_6_ and (**c**) La-doped Bi_2_MoO_6_ nanocomposites and of deconvoluted Mo 4f region of (**b**) Bi_2_MoO_6_ and (**d**) La-doped Bi_2_MoO_6_ nanocomposites.

**Figure 8 materials-13-00035-f008:**
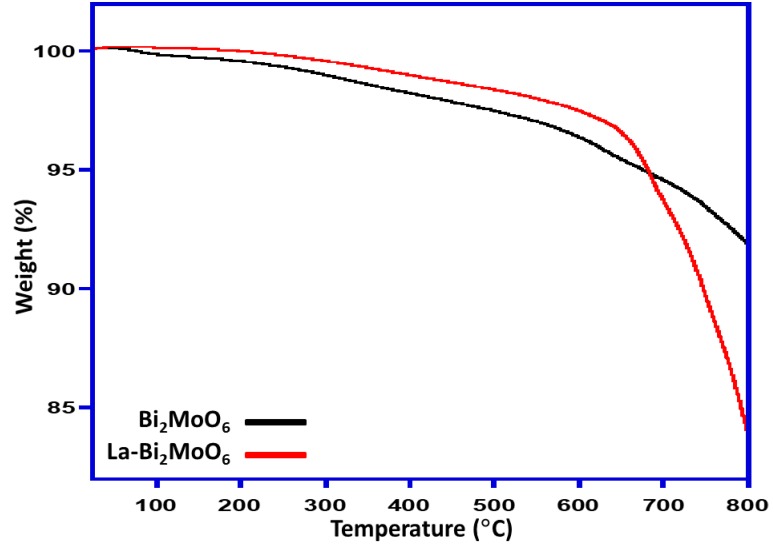
Thermal degradation patterns of Bi_2_MoO_6_ and La-doped Bi_2_MoO_6_ nanocomposites.

**Figure 9 materials-13-00035-f009:**
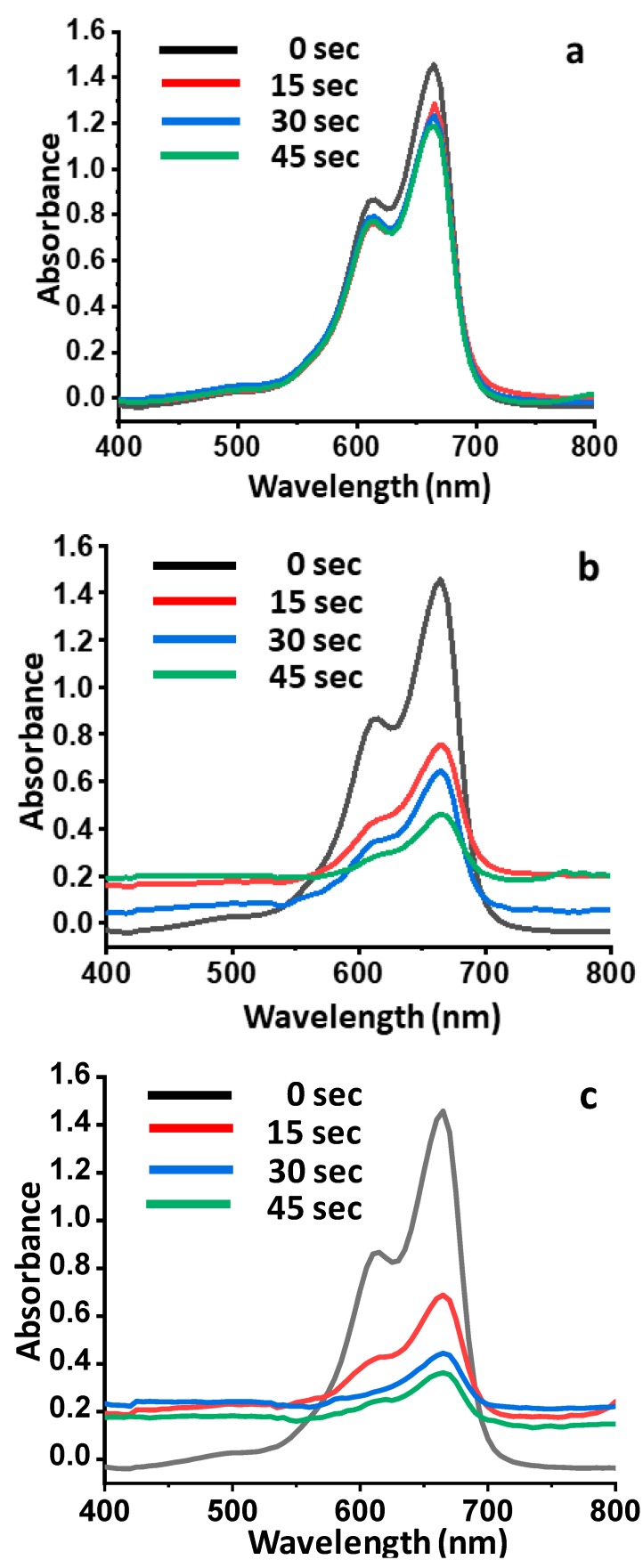
(**a**) UV–VIS spectra showing the degradation of methylene blue using NaBH_4_. (**b**) UV–VIS spectra showing the degradation of methylene blue using NaBH_4_ in the presence of Bi_2_MoO_6_. (**c**) UV–VIS spectra showing the degradation of methylene blue using NaBH_4_ in the presence of La-doped Bi_2_MoO_6._

**Figure 10 materials-13-00035-f010:**
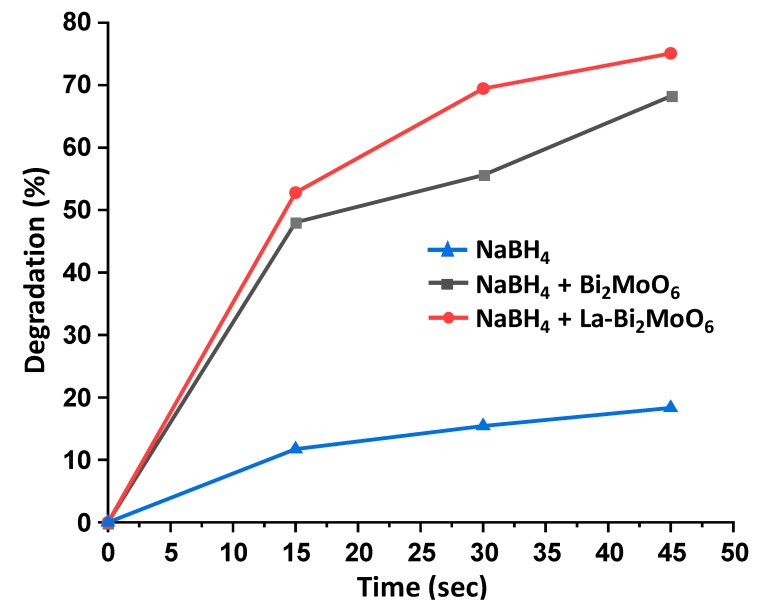
Graphical illustration of the photocatalytic degradation kinetics of methylene blue using NaBH_4_, NaBH_4_ in the presence of Bi_2_MoO_6_, and NaBH_4_ in the presence of La-doped Bi_2_MoO_6._

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
