# Peer review of "Enhanced Photoluminescence and Photocatalytic Efficiency of La-Doped Bismuth Molybdate: Its Preparation and Characterization"

_materials, 2019, doi:10.3390/ma13010035_

Round 1

Reviewer 1 Report

Muhammad Waqar and co-authors present a systematic work on the synthesis, characterization, and application of lanthanide doped (La-doped) bismuth molybdate (Bi2MoO6) materials. The obtained results showed and improved photocatalytic properties. The manuscript is well prepared and organized.  I recommend the publication of this work after major revision:

The size of the particle from SEM/TEM is not the same with those obtained from XRD, can the authors check the measurements. The section of Photo-catalytic activity should be completely rewritten; (a) the discussion of photocatalytic results is not clear and the manner to present the performance of the materials is not visible (b) the kinetic study must be separated in other title (c) to demonstrate the efficiency of the materials a comparison with other published work must be added.

Author Response

Muhammad Waqar and co-authors present a systematic work on the synthesis, characterization, and application of lanthanide doped (La-doped) bismuth molybdate (Bi2MoO6) materials. The obtained results showed and improved photocatalytic properties. The manuscript is well prepared and organized.  I recommend the publication of this work after major revision:

Comment: The size of the particle from SEM/TEM is not the same with those obtained from XRD, can the authors check the measurements.

Response: Measurements and calculations have been recalculated and corrected. Moreover, we would like to mention that from XRD the crystallite size was calculated, which is different from the grain size calculated from the SEM/TEM images, hence it is not necessary that both the values obtained from SEM and XRD data be the same. [Kanwal, Farah, et al. "Synthesis of polypyrrole–ferric oxide (Ppy–Fe2o3) composites and study of their structural and conducting properties." Synthetic Metals 161.3-4 (2011): 335-339.]

Comment: The section of Photo-catalytic activity should be completely rewritten; (a) the discussion of photocatalytic results is not clear and the manner to present the performance of the materials is not visible (b) the kinetic study must be separated in another title (c) to demonstrate the efficiency of the materials a comparison with other published work must be added. 

Response: The photocatalytic activity section has been rewritten as advised by the referee keeping the viewpoints (a-c) into consideration.

Reviewer 2 Report

This is an interesting article, it is dealing systematic study of the enhanced physicochemical properties of lanthanide doped (La-doped) bismuth molybdate (Bi2MoO6) is performed.

Some comments:

In the abstract and in the experimental section, in the chemical compounds Na2MoO4.2H2O, LaCl3.7H2O, dots are inserted instead of multiplication.  It is necessary for example, LaCl3·7H2O Please check the references throughout the text. For example, [8] - [9] [10-12] is most likely here [8-12] or [13]-[12, 14, 15] probably here [12- 15] etc... line 201: [The Development of Pseudocapacitive Molybdenum Oxynitride Electrodes for Super capacitors, Haoran Wu and Keryn Lian, ECS Transactions, 58 (25) 67-75 (2014)] most likely needs to be moved to the References section. line 208: What does subsection 3.1.2 belong to?

Author Response

This is an interesting article; it is dealing systematic study of the enhanced physicochemical properties of lanthanide doped (La-doped) bismuth molybdate (Bi2MoO6) is performed.

Some comments:

Comment: In the abstract and in the experimental section, in the chemical compounds Na2MoO4.2H2O, LaCl3.7H2O, dots are inserted instead of multiplication.  It is necessary for example, LaCl3·7H2O

Response: The error pointed out has been corrected as suggested by the referee.

Comment: Please check the references throughout the text.

For example, [8] - [9] [10-12] is most likely here [8-12] or [13]-[12, 14, 15] probably here [12- 15] etc...

Response: References have been merged as [8-12] and [13-15] as pointed out by the referee.

line 201: [The Development of Pseudocapacitive Molybdenum Oxynitride Electrodes for Supercapacitors, Haoran Wu and Keryn Lian, ECS Transactions, 58 (25) 67-75 (2014)] most likely needs to be moved to the References section.

Response: This reference has been moved to the reference section as reference no. 29

Comment: line 208: What does subsection 3.1.2 belong to? 

Response: line 208: This was a typographical error and hence subsection numbers have been removed throughout the manuscript.

Reviewer 3 Report

In this work, the authors propose a study of the chemical and physical properties of Bi2MoO6 (BMO), undoped and doped by La. They study also the photocatalytic activity of the two samples in order to evaluate the enhanced potentiality of La-dopes material for future uses in water splitting and in industrial applications, such as for the dyes degradation under visible-light irradiation.

After a structural and morphological analysis (XRD, BET, TGA…), they add a spectroscopic investigation by luminescence and absorption measurement. The methylene blue MB degradation in presence of a reducing agent, (dye is chosen as reference) is calculated and given as % reduction efficiency. La doped samples show a greater catalytic activity with respect to the undoped one.

The topic and materials under investigation are attractive and several related papers have been published recently (such as Zhang, P., Yi, Y., Yu, C. et al. J Mater Sci: Mater Electron (2018) 29: 8617, or Mu, J.J., Zheng, G.H., Dai, Z.X. et al. J Mater Sci: Mater Electron (2017) 28: 14747). According to the literature, this work do not provide a step forward in the development of efficient materials for photo-catalysis, as the results that the authors propose trace some already publications.

Moreover, I am convinced that this work needs major revisions before the publication, and only after these revisions, it might be accepted.

1.      Style/punctuation/ English grammar must be corrected.

2.      Some articles are absent before the corresponding words and others are superfluous.

3.      The authors call BiMoO6 “precursor”, I suggest to use “undoped”

4.      Style of the figures is not uniform.

5.      Pag 2 line 56: when the authors say “as it is well documented in the literature..” they should provide the references they are referring to

6.      Figure 4 FT-IR legend table is missing.

7.      Pag 6 Line180 Please correct Figure XXX by putting the correct number

8.      Pag 6 Line192 “…successful doping of La in the nanocomposite given in fig 6. I suppose it is figure 7. Pag 8 Line 203 Please check the number of figure (and while doing this, please check the correct correspondence for all the figures)

9.      In the conclusions, the authors say that the La doped Bi2MoO6 has greater performances than undoped sample and it can be due to the structural transformation occurring during the preparation. Can the authors give clarifications? It is known that the substitution doping of rare earth element in Bi2MoO6 significantly influences the morphology of the host materials. Can you find this evidence in your investigation? I suppose yes, according to TGA, XRD and BET…Also, how do the volume/dimension of the pores influence the ability of the materials in promoting the separation efficiency of the electron–hole pairs? And the shape of the materials?

10.  Consider the role of traps in RE doped materials to prevent the recombination of e−/h+

11.  What do the authors mean @ pag 3with Figure 1 shows the SEM micrograms of the prepared Bi2MoO6 and La-doped Bi2MoO6 nanocomposite. The Bi2MoO6 surface morphology appears to be dense rod-like structures, however, upon doping the Bi2MoO6 with La, the surface morphology is found to be modified to scattered

La increases the roughness of the surface? Or induces agglomeration?

12.  Fig2: the average size is 3.5 nm in both the samples, just the number of nanopart. with diameter >5 nm is greater in the La doped BMO

13.  Figure 3 is not clear. It is not easy to recognize a distribution here... How have the authors obtained the distribution depicted? And the determination of the pore size? Please give details. Why do the author consider only the Min/Max values to five the pore volume range of the samples? The dimensions of the pores are of the “granular”/agglomerations or of the small nanopart.?

14.  FT-IR , please say briefly why the shift is revealing the incorporation of La, or give a references

15.  XRD patterns results five diameters of 17.57-95.51 nm of the particle. What about the comparison with TEM? Are they the same materials?

16.  Luminescence in not clear. Do the authors say that in BMO and La-BMO the 1S0-3P1 transition is partially allowed due to the spin orbit coupling? Therefore, the absorption should be from 300-350 nm. Can the authors provide it? I cannot understand the origin of the 700 nm emission and why the increase of this in La doped sample is related to the increase in photo-catalytic activity. Is figure 6 representing the PL spectra just of the BMO and La-BMO powders?

17.  Figure 8, I suggest to mark the peaks with the corresp. element

18.  I suggest adding a figure where the degradation of the dye MB is represented as a function of time. By using the equation at pag 3, the authors can plot C/Co vs time: in this way a clear interpretation of the results is provided to the readers

19.  In general, the work "has a structure", nevertheless the results are not well described and the conclusions/deductions coming from them are not presented.  I suggest a clarification/refinement of the presentation of the results in order to highlight what is the take home message of the work.

Author Response

In this work, the authors propose a study of the chemical and physical properties of Bi2MoO6 (BMO), undoped and doped by La. They study also the photocatalytic activity of the two samples in order to evaluate the enhanced potentiality of La-dopes material for future uses in water splitting and in industrial applications, such as for the dyes degradation under visible-light irradiation.

 After a structural and morphological analysis (XRD, BET, TGA…), they add a spectroscopic investigation by luminescence and absorption measurement. The methylene blue MB degradation in presence of a reducing agent, (dye is chosen as reference) is calculated and given as % reduction efficiency. La doped samples show a greater catalytic activity with respect to the undoped one.

The topic and materials under investigation are attractive and several related papers have been published recently (such as Zhang, P., Yi, Y., Yu, C. et al. J Mater Sci: Mater Electron (2018) 29: 8617, or Mu, J.J., Zheng, G.H., Dai, Z.X. et al. J Mater Sci: Mater Electron (2017) 28: 14747). According to the literature, this work do not provide a step forward in the development of efficient materials for photo-catalysis, as the results that the authors propose trace some already publications.

Moreover, I am convinced that this work needs major revisions before the publication, and only after these revisions, it might be accepted.

Comment: Style/punctuation/ English grammar must be corrected.

Response: Style/punctuation/ English grammar has been corrected by the proofreading of the manuscript with utmost care.

Comment: Some articles are absent before the corresponding words and others are superfluous.

Response: Necessary ‘articles’ have been added before the corresponding words and others have been removed as pointed out by the referee.

Comment: The authors call BiMoO6 “precursor”, I suggest to use “undoped”

Response: Change suggested by the referee is appreciated and has been implemented throughout the manuscript.

Comment:  Style of the figures is not uniform.

Response: Indeed, we agree that the figures are not a uniform style, especially the FT-IR spectra presented. However, since we are not having the RAW data of the spectra’s reported and in order to get things in uniformity, we have decided to move the figure to the Supplementary file.

Comment: Pag 2 line 56: when the authors say “as it is well documented in the literature..” they should provide the references they are referring to

Response: Relevant literature has been added as pointed out by the referee

Comment: Figure 4 FT-IR legend table is missing.

Response: The figure is labeled as a) and b) and this is explained in the figure caption.

Comment:  Pag 6 Line180 Please correct Figure XXX by putting the correct number

Response: Figure 6 has been mentioned as pointed out by the referee.

Comment:   Pag 6 Line192 “…successful doping of La in the nanocomposite given in fig 6. I suppose it is figure 7. Pag 8 Line 203 Please check the number of figure (and while doing this, please check the correct correspondence for all the figures)

Response: Yes, it is fig 7 and has been corrected, in continuation to this all figures numbers have been rechecked.

Comment: In the conclusions, the authors say that the La doped Bi2MoO6 has greater performances than undoped sample and it can be due to the structural transformation occurring during the preparation. Can the authors give clarifications? It is known that the substitution doping of rare earth element in Bi2MoO6 significantly influences the morphology of the host materials. Can you find this evidence in your investigation? I suppose yes, according to TGA, XRD and BET…Also, how do the volume/dimension of the pores influence the ability of the materials in promoting the separation efficiency of the electron–hole pairs? And the shape of the materials?

Response: The conclusion part has been revised to remove ambiguity as pointed out by the referee.

Comment: Consider the role of traps in RE doped materials to prevent the recombination of e−/h+

Response: The trapped electrons are released slowly and recombine with the recombination center to generate La3+ emission. Most probably, the mechanism is due to the electron trapping and electron release. We are doing a series of work and in upcoming contributions, we will be focusing to consider the role of traps as suggested by referee.

Comment: 11.  What do the authors mean @ pag 3 with Figure 1 shows the SEM micrograms of the prepared Bi2MoO6 and La-doped Bi2MoO6 nanocomposite. The Bi2MoO6 surface morphology appears to be dense rod-like structures, however, upon doping the Bi2MoO6 with La, the surface morphology is found to be modified to scattered

Response: Probably in the present case, it seems to increases the roughness of the surface due to coarse granular clustering, however, the phrase has also been modified in the text.

Comment: Fig2: the average size is 3.5 nm in both the samples, just the number of nanopart. with diameter >0.5 nm is greater in the La doped BMO

Response: Yes, it appears to be a minor difference in grain size, however, the values given are from the values obtained from the Gaussian fitting of the particle distribution graph.

Comment:  Figure 3 is not clear. It is not easy to recognize a distribution here... How have the authors obtained the distribution depicted? And the determination of the pore size? Please give details. Why do the author consider only the Min/Max values to five the pore volume range of the samples? The dimensions of the pores are of the “granular”/agglomerations or of the small nanopart.?

Response: The distribution depicted, the pore size and volume were obtained by the BET analysis. The section has been modified with relevant references have been added.

Comment: FT-IR , please say briefly why the shift is revealing the incorporation of La, or give a references

Response: Relevant reference has been added.

Comment:  XRD patterns results five diameters of 17.57-95.51 nm of the particle. What about the comparison with TEM? Are they the same materials?

Response: Measurements and calculations have been recalculated and corrected. Moreover, we would like to mention that from XRD the crystallite size was calculated, which is different from the grain size calculated from the SEM/TEM images, hence it is not necessary that both the values obtained from SEM and XRD data be the same. [Kanwal, Farah, et al. "Synthesis of polypyrrole–ferric oxide (Ppy–Fe2o3) composites and study of their structural and conducting properties." Synthetic Metals 161.3-4 (2011): 335-339.]

Comment:  Luminescence in not clear. Do the authors say that in BMO and La-BMO the 1S0-3P1 transition is partially allowed due to the spin orbit coupling? Therefore, the absorption should be from 300-350 nm. Can the authors provide it? I cannot understand the origin of the 700 nm emission and why the increase of this in La doped sample is related to the increase in photo-catalytic activity. Is figure 6 representing the PL spectra just of the BMO and La-BMO powders?

Response: This part has been revised with the support of previous literature as suggested by the referee.

Comment: I suggest adding a figure where the degradation of the dye MB is represented as a function of time. By using the equation at pag 3, the authors can plot C/Co vs time: in this way a clear interpretation of the results is provided to the readers

Response: This figure has been added for clarity to readers as suggested by the referee.

Comment:  In general, the work "has a structure", nevertheless the results are not well described and the conclusions/deductions coming from them are not presented.  I suggest a clarification/refinement of the presentation of the results in order to highlight what is the take-home message of the work.

Response: Results have been presented and discussed with literature. Hopefully revised form will fulfill all points raised by the referee.

Round 2

Reviewer 3 Report

The authors propose a corrected and updated version of the manuscript. Details and new measurements are presented.

Just 1 thing needs to be checked.

XRD and TEM size difference. The authors say (and I agree) that the 2 methods can give the same results as well as different results (Langmuir 2005, 21, 1931-1936 ). Still, I would like to know why in this case they are different (1 order of magnitude: from 3 nm TEM to more then 10 nm XRD) What is attributed to grains and what to crystallite size.

After this clarification, the paper can be published

Author Response

The authors propose a corrected and updated version of the manuscript. Details and new measurements are presented.

Just 1 thing needs to be checked.

Comment: XRD and TEM size difference. The authors say (and I agree) that the 2 methods can give the same results as well as different results (Langmuir 2005, 21, 1931-1936 ). Still, I would like to know why in this case they are different (1 order of magnitude: from 3 nm TEM to more then 10 nm XRD) What is attributed to grains and what to crystallite size.

Response: We thank the reviewer for pointing this observed discrepancy. We have provided the following response in the revised manuscript. Normally the size of crystallites as obtained by XRD are smaller than the particle size obtained by TEM, as the particles are made up of crystallites. However, in our case the observation is opposite, this may be because XRD measurements are usually on bulk of the sample, whereas TEM is usually on a small selected area, which in our case showed smaller particle size. (line 166-169 revised manuscript)